# Polypyrrole Nanosheets Prepared by Rapid In Situ Polymerization for NIR-II Photoacoustic-Guided Photothermal Tumor Therapy

Yixin Xie [1,†], Ji Xu [1,†], Hui Jin [1,†], Yunfeng Yi [2], Yuqing Shen [3], Xiuming Zhang [1], Xinxin Liu [1], Yanan Sun [1], Wei Shi [1,*], Yuan He [2,*] and Dongtao Ge [1,*]

[1]   The Higher Educational Key Laboratory for Biomedical Engineering of Fujian Province, Research Center of Biomedical Engineering of Xiamen, Fujian Key Laboratory of Surface and Interface Engineering for High Performance Materials, Department of Biomaterials, College of Materials, Xiamen University, Xiamen 361001, China; 20720201150140@stu.xmu.edu.cn (Y.X.); 31420191150146@stu.xmu.edu.cn (J.X.); 31420181150116@stu.xmu.edu.cn (H.J.); zhangxiuming@xmu.edu.cn (X.Z.); liuxinxin@xmu.edu.cn (X.L.); sunyanan@xmu.edu.cn (Y.S.)

[2]   Department of Cardiothoracic Surgery, The 909th Hospital, School of Medicine, Xiamen University, Zhangzhou 363000, China; yyfeng.dor1969@163.com

[3]   Transfusion Department, Woman and Children's Hospital, School of Medicine, Xiamen University, Xiamen 361001, China; syuqing@163.com

*   Correspondence: shiwei@xmu.edu.cn (W.S.); heyuan_1988@126.com (Y.H.); gedt@xmu.edu.cn (D.G.)

†   These authors contributed equally to this work.

**Abstract:** Recently, the near-infrared-II (NIR-II, 1000–1350 nm) region has been extensively applied in deep-tissue photothermal therapy (PTT) on account of it having stronger tissue penetration and a higher maximum permissible exposure (MPE) than the near-infrared-I (NIR-I, 650–950 nm) region. In this study, we developed a rapid and convenient in situ polymerization strategy to fabricate polypyrrole nanosheets (PPy NSs) within a few minutes using manganese dioxide nanosheets ($MnO_2$ NSs) as both the oxidant and the self-sacrificed template. The fabricated PPy NSs exhibited excellent NIR-II absorption, which conferred its high photothermal conversion efficiency (66.01%) at 1064 nm and its photoacoustic (PA) imaging capability. Both in vivo and in vitro studies have shown that that PPy NSs possess good biological safety and excellent PTT efficacy and PA imaging performances. Thus, the as-synthesized PPy NSs could effectively achieve PA imaging-guided photothermal tumor ablation under 1064 nm excitation. Our work provides a novel and promising method for the rapid preparation of PPy NSs without the addition of exogenous oxidants and subsequent template removal, which could be regarded as potential photothermal agents (PTAs) to integrate the diagnosis and treatment of cancer.

**Keywords:** polypyrrole nanosheets; in situ polymerization; near-infrared-II region; photoacoustic imaging; photothermal therapy





## 1. Introduction

Photothermal therapy (PTT) is a new and promising cancer therapeutic modality that utilizes photothermal agents (PTAs) to achieve effective photo-to-thermal conversion to induce localized high temperature (>43 °C) for the ablation of tumor tissue [1–3]. The tumor targeted enrichment of nano PTAs and localized irradiation can ensure that the generated high heat is confined to the tumor area, thus effectively minimizing thermal damage to surrounding normal tissues. Compared with clinical cancer treatments, such as chemotherapy, radiotherapy, and surgical intervention, PTT has the advantages of high selectivity, low toxicity, low invasiveness, low dependency on the microenvironment, and a short curative time [4]. Additionally, PTT could enhance the sensitivity of tumor cells to

radiation and chemotherapy as well [5,6]. Ideal PTAs not only require good biocompatibility, but also possess strong light absorption, especially in the near-infrared (NIR) region, a high photothermal conversion property, and photostability [7,8]. Therefore, developing appropriate PTAs for clinical photothermal therapy remains a major challenge in achieving highly efficient PTT. With the integrated development of materials science and biomedicine, researchers have developed a variety of inorganic [9,10] and organic functional [11,12] nanomaterials as PTAs applicable for cancer photothermal diagnosis and treatment, such as noble metal [13–15], sulfides [16,17], transition metal dichalcogenides [18,19], polypyrrole (PPy) [20–22], polydopamine [15,23], and indocyanine green [24,25], which have been reported as having an excellent PTT effect for cancer treatment.

However, most PTAs are only responsive to light in the NIR-I region, which has a weaker tissue-penetration depth and higher tissue scattering and reabsorption, as well as a lower maximum allowable exposure threshold compared to the NIR-II optical window [26,27]. Therefore, developing PTAs with good photothermal performance in the NIR-II region has high value in respect of its clinical application. In fact, the light absorption properties of conjugated polymers could be enhanced by regulating the degree of polymerization and morphology [28]. Compared with zero-dimensional nanospheres, two-dimensional nanosheets have a larger specific surface area and unique optical properties, and many researchers have reported 2D nanosheets with excellent NIR-II responses [10,29].

PPy as a typical conducting polymer has excellent biocompatibility [30,31] and photostability, good tunable broadband absorption, and good photothermal conversion performance. Meanwhile, polypyrrole has also been shown to have certain photoacoustic imaging properties. Photoacoustic imaging (PAI) is a new imaging mode, which can obtain optical contrast in the tissue. It has wide application value in the early diagnosis of cancer and in certain other disease-detection procedures and scientific research. PPy is an excellent photoacoustic imaging contrast agent. It has more conjugated bonds and can obtain a high photoacoustic signal. Through these properties, PPy nanoparticles can combine photoacoustic imaging with photothermal imaging in vivo to obtain better therapeutic and diagnostic effects [32,33]. At present, there exist several studies on the application of PPy in the NIR-II region PTT [21,34]. However, the greatest problem in the preparation of PPy nanomaterials is the poor controllability of polymerization and morphology. Almost all reported methods for the fabrication of PPy nanostructures including nanospheres (NPs) [35], nanotubes (NTs) [36], and nanosheets (NSs) [36–38] need the assistance of templates (soft or hard templates). Meanwhile, difficult template removal further leads to fewer reports on the application of pure PPy NSs PATs in biomedicine. Xu et al. first successfully prepared ultrathin PPy NSs using layered FeOCl as a hard space-confined template, and the PPy NSs showed excellent NIR-II photothermal capacity [36]. Cai et al. reported efficient photothermal performance from hexagonal polypyrrole (PPy) NSs that were prepared by means of complex and fine interfacial polymerization [37]. Im et al. fabricated PPy NSs via an organic crystal surface-induced method in which the hydrated crystal of sodium decylsulfonate was used as the template and $FeCl_3$ as the oxidant [38]. Apetrei, RM, et al. used a whole-cell culture and cell-free crude enzyme extract achieving the cell-assisted enzymatic polymerization/oligomerization of PPy [39]. However, these methods still have the problems of complex preparation operations, long reaction times, and a cumbersome post-purification process. Therefore, it is necessary to develop fast and simple PPy synthesis methods.

Herein, we proposed a convenient and rapid in situ polymerization strategy for the preparation of PPy NSs, in which the polymerization of pyrrole took only approximately 2 min without additional oxidant and a template removal process. In this method, manganese dioxide nanosheets ($MnO_2$ NSs) acted as both the spatial 2D template and the oxidant to directly guide the formation of PPy NSs. Notably, the polymerization of PPy NSs could be achieved within only 2 min, accompanied by the dissolution and consumption of $MnO_2$. The prepared PPy NSs exhibited good photothermal and photoacoustic (PA) imaging capabilities in the NIR-II region. The photothermal conversion efficiency

(PTCE) of PPy NSs could reach up to 66.01% at 1064 nm. After modifying with DSPE-PEG, PPy@DSPE-PEG NSs possessed better dispersion stability and biological safety. More importantly, in vitro and in vivo experiments showed that PPy@DSPE-PEG NSs could effectively achieve cancer ablation via PA imaging-guided PTT in the NIR-II region (Figure 1).

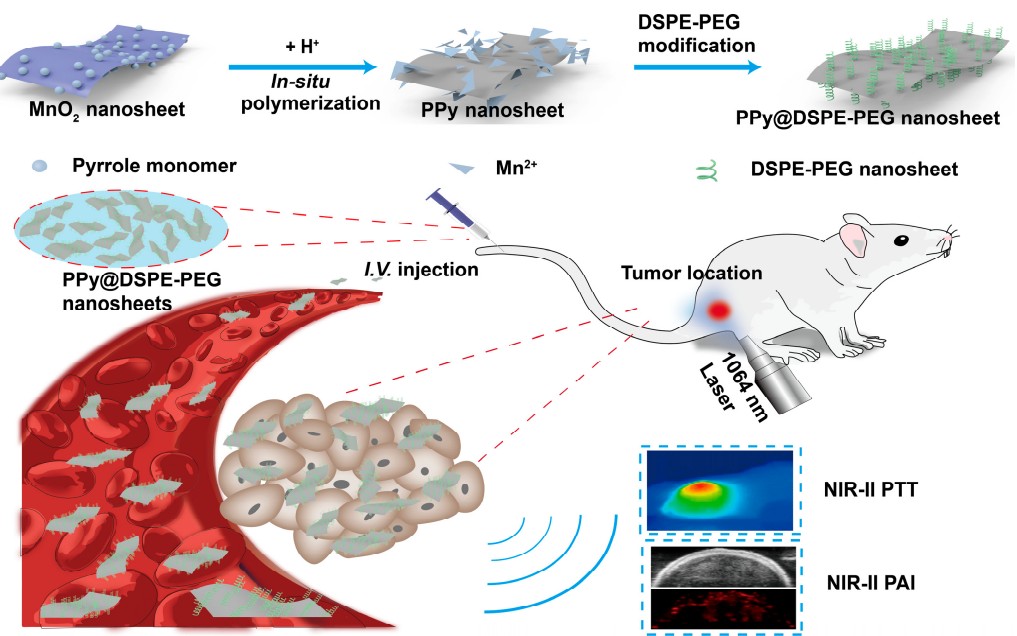

**Figure 1.** Schematic of synthetic route of PPy NSs via in situ polymerization and in vivo photoacoustic imaging-guided photothermal therapy.

## 2. Materials and Methods

### 2.1. Materials

Tetramethylammonium hydroxide ($C_4H_{13}NO$), manganese chloride tetrahydrate ($MnCl_2 \cdot 4H_2O$), methoxyl poly (ethylene glycol)-phosphatidylethanolamine (mPEG-DSPE, Mw: 2000), and sodium dodecyl sulfate (SDS) were purchased from Aladdin Industrial Corporation (Shanghai, China). Hydrogen peroxide ($H_2O_2$, 30%), methanol ($CH_3OH$), and hydrochloric acid (HCl, 36%) were purchased from Beijing SINOPHARM (Beijing, China). Fetal bovine serum (FBS), DMEM, and 1640 medium were purchased from Gibco (Grand Island, NE, USA). The above reagents were directly used without further purification. Pyrrole (98%) was purchased from Beijing J&K Chemical Ltd. (Beijing, China) and purified by distillation under the protection of $N_2$ gas and stored in the refrigerator at $-20\ ^\circ$C for future use. Ultrapure water (18.2 MΩ) was used in all experiments.

### 2.2. Synthesis of MnO₂ NSs

$MnO_2$ NSs were synthesized according to the reported study [40,41]. In brief, 20 mL fresh aqueous solution containing 0.6 M $C_4H_{13}NO$ and 3 wt.% $H_2O_2$ was rapidly added into the $MnCl_2$ (10 mL, 0.3 M) solution within 15 s under vigorous stirring and further reacted for 12 h at room temperature. Then, the precipitates of reduced crude $MnO_2$ were collected by centrifugation (2000 rpm, 20 min) and washed several times with ultrapure water and methanol, respectively. Next, the $MnO_2$ precipitates were dried in an oven at 60 $^\circ$C. Finally, 10 mg of $MnO_2$ was ground and dispersed in 20 mL ultrapure water and ultrasonicated (power: 480 W) for 12 h to strip the $MnO_2$ NSs. Large particles were removed by centrifugation at 2000 rpm for 30 min, and the $MnO_2$ NSs in the supernatant were collected.

### 2.3. Synthesis of PPy NSs and PPy@DSPE-PEG NSs

PPy NSs were synthesized by means of the in situ oxidation of the $MnO_2$ NSs. Firstly, 15 mL pyrrole aqueous solution containing 100 μL pyrrole and 0.01 g SDS were added into the $MnO_2$ NS aqueous solution (15 mL, 50 μg/mL) and further magnetically stirred for 20 min in an ice water bath. Then, 1 mL HCl (0.01 M) was rapidly injected into the mixture under stirring, and the reaction continued for approximately 2 min with the solution color changing quickly from brown–yellow to transparent gray–black. The excess SDS, PY, and reduced $Mn^{2+}$ ions were removed by washing in a 50 mL ultrafiltration tube (30 kDa) with ultra-pure water, and the PPy NSs were collected. Finally, PPy NSs and mPEG-DSPE were mixed at a mass ratio of 1:10 by means of ultrasonication for 5 min to prepare the PPy@DSPE-PEG NSs.

### 2.4. Characterization

The morphologies of $MnO_2$ and PPy NSs were performed by using scanning electron microscopy (SEM, Hitachi, SU-70, Tokyo, Japan), transmission electron microscopy (TEM, FEI, Talos F200s, Houston, TX, USA) at an accelerating voltage of 200 kV and atomic force microscopy (AFM, BRUKER, Multimode 8, Billerica, MA, USA) at Multimode 8 mode. The chemical and structural composition of the NSs were characterized by performing Fourier transform infrared spectroscopy (FTIR, Thermo Fisher, Nicolet Is10, Waltham, MA, USA), UV–Vis–NIR spectrometry (SHIMADZU, UV-3600+, Kyoto, Japan), and X-ray diffraction (XRD, BRUKER, D8-A25, USA). The Malvern Zetasizer Nano ZS analyzer (Malvern, Mastersizer2000/MAL1012737, Shanghai, China) was used to measure the hydrodynamic diameter and zeta potential of nanomaterials by dynamic light scattering (DLS).

### 2.5. Photothermal Effect

In order to explore the photothermal effect caused by NIR irradiation, the aqueous solutions of PPy@DSPE-PEG NSs with various concentrations (0, 20, 50, 100, 200, and 300 μg/mL) were irradiated with an 808/1064 nm NIR laser (Beijing Hi-tech Optoelectronics Co., Ltd., Beijing, China) at 1 W/cm² for 5 min. Meanwhile, the temperature of the solutions was recorded by an infrared thermal camera (FOTRIC, FOTRIC-225, Shanghai, China). To measure the photothermal stability, 100 μg/mL PPy@DSPE-PEG NS solution was exposed to the 1064 nm laser at 1 W/cm² for 300 s and then was cooled to room temperature. The heating and cooling cycles were repeated five times.

To test the photothermal effect of the PPy@DSPE-PEG NSs under the cover skin condition, quartz cuvettes containing 100 μg/mL PPy@DSPE-PEG NSs aqueous solution were covered with chicken breasts of different thicknesses (0, 2, 4, 6, 8, and 10 mm). Meanwhile, the thermal imaging system was used to record the temperature rise of the solution under the 808/1064 nm laser irradiation (1 W/cm², 5 min).

The photothermal conversion efficiency of the PPy@DSPE-PEG NSs was calculated by using the following formula:

$$\eta = \frac{hA\Delta T_{max} - Q_s}{I(1 - 10^{-A_\lambda})} \tag{1}$$

Among these, $hA = mC/\tau_s$, where m is the mass of water, C is the specific heat capacity of water, and τs is the cooling time constant of the PPy nanosheet. $\Delta T_{max} = T_{max} - T_{surr}$, which means the difference between the maximum heating temperature of the PPy nanosheet and room temperature. $Q_s = mC \cdot \Delta T_{water}$, which is the heat absorbed by water. *I* is the power density of the laser. $A_\lambda$ is the absorption value of the PPy nanosheet at different wavelengths.

### 2.6. Cytotoxicity Assay of PPy@DSPE-PEG NSs

A mouse fibroblast cell line (L929) and a mouse breast cancer cell line (4T1) were obtained from the China Center for Type Culture Collection (CCTCC). In a 37 °C, 5% $CO_2$ incubator atmosphere, L929 and 4T1 cells were cultured in 1640 and DMEM medium, respectively, both supplemented with 10% fetal bovine serum (FBS). Briefly, L929 and

4T1 cells were seeded into 96-well plates at a density of $1 \times 10^4$ cells/well. After 24 h of incubation, the medium was replaced by fresh complete medium containing PPy@DSPE-PEG NSs with different concentrations (0, 20, 50, 100, and 400 μg/mL). Then, the MTT assay ($n = 5$) was used to evaluate the cell viability by measuring the optical density at 490 nm and 570 nm with a microplate reader (Tecan, Infinity 200 pro, Mennedorf, Switzerland).

### 2.7. In Vitro Photothermal Therapy

The photothermal cytotoxicity of the PPy@DSPE-PEG NSs was evaluated on 4T1 cells. The 4T1 cells were cultured in 24-well plates for 24 h at 37 °C; then, different concentrations of PPy@DSPE-PEG NSs were added to the culture dish for co-incubation with the cells. Subsequently, the PPy@DSPE-PEG NSs were irradiated with an 808/1064 nm laser at a power density of 1 W/cm$^2$ for 10 min, and cell viability was observed for 24 h. The viability of 4T1 cells was evaluated by using MTT ($n = 5$) and a calcein AM/propidium iodide (PI) reagent. First, 5 μL 16 mM PI was added into 10 mL PBS; then, 5 μL 4 mM AM was added into it to prepare the working solution. Then, 1 mL working solution was pipetted into every well, followed by incubation for 30 min. Finally, the samples were visualized using a fluorescence microscope (Leica, Leica DMi8, Wetzlar, Germany).

### 2.8. Tumor Model

Female BALB/c mice (6~8 weeks) used in the experiment were purchased from Shanghai Slack Laboratory Animal Co., Ltd., Shanghai, China. All animal experiments were performed under the guidelines outlined in the Guide for the Care and Use of Laboratory Animals. All procedures were approved by the Experimental Animal Center Committee of Xiamen University. A quantity of 100 μL of 4T1 cells with a density of $1 \times 10^7$ cells/mL was subcutaneously injected into the right hind leg of female BALB/c mice. When the tumor volume reached 100 mm$^3$, the 4T1 tumor-bearing mice were used for subsequent experiments. The tumor volume calculation formula is (tumor length) $\times$ (tumor width)$^2$ $\times$ 0.5.

### 2.9. In Vivo Photoacoustic Imaging

The PA imaging of the PPy@DSPE-PEG NSs was evaluated by using the Multi-mode Ultrasound/Photoacoustic Imaging System (FUJIFILM Visual Sonics, VEVO LAZR-X, Bothell, WA, USA). The 4T1 tumor-bearing BALB/c mice were injected intravenously with PPy@DSPE-PEG NSs at 4 mg/kg to study in vivo photoacoustic imaging. The photoacoustic signal of tumor region was detected at a wavelength of 1064 nm after injection for 2, 4, 8, 12, and 24 h.

### 2.10. In Vivo Photothermal Therapy

4T1 tumor-bearing mice were randomly divided into six groups ($n = 5$ per group) and marked as (1) PBS; (2) PPy; (3) PPy/0.5 mm, 808 nm; (4) PPy/0.5 mm, 1064 nm; (5) PPy, 808 nm; and (6) PPy, 1064 nm. In brief, the control group of mice was injected intravenously with PBS; the other five experimental groups were injected intravenously with the same dose of 4 mg/kg PPy@DSPE-PEG NSs dispersed in PBS. At 8 h after the injection, mice in four laser irradiation treatment groups were irradiated with the corresponding 808/1064 nm laser (1 W/cm$^2$) for 10 min; two of these groups were also irradiated covered with 0.5 mm chicken breast. In the first three days of the experiment, they were irradiated continuously, and the real-time temperature of the tumor region was recorded by the infrared thermal camera. The tumor size and body weight of the mice were measured every 2 days. At the 14th day of treatment, all the mice were sacrificed, and their major organs (heart, liver, spleen, lung, and kidney) and tumors were harvested for H&E staining to analyze the biological safety.

## 3. Results and Discussion

### 3.1. Synthesis and Characterization of PPy NSs

The synthesis process of PPy NSs is schematically illustrated in Figure 2a. To obtain PPy NSs, ultra-thin $MnO_2$ NSs as the self-sacrificial spatial oxidant template needed to be prepared first. The $MnO_2$ NS ultra-thin structure was prepared by using an ultrasonic stripping method [42]. SEM and TEM images of stripped $MnO_2$ NSs showed obviously lamellar nanostructures with clear shadow fold stripes (Figure 2b and Figure S1a, ESI†). The measured thickness of the $MnO_2$ NSs was approximately 1~2 nm, and the size was ~150 nm as shown by AFM observation (Figure S1b, ESI†). The UV–Vis spectrum of $MnO2$ NSs exhibited only one characteristic absorption peak at 374 nm and had almost no absorption in the NIR region (Figure 2c). The diffraction patterns in the XRD spectrum were mainly at 12.6°, 25°, and 37.1°, corresponding to the (001), (002), and (110) planes of $MnO_2$ (Figure 2d) [43].

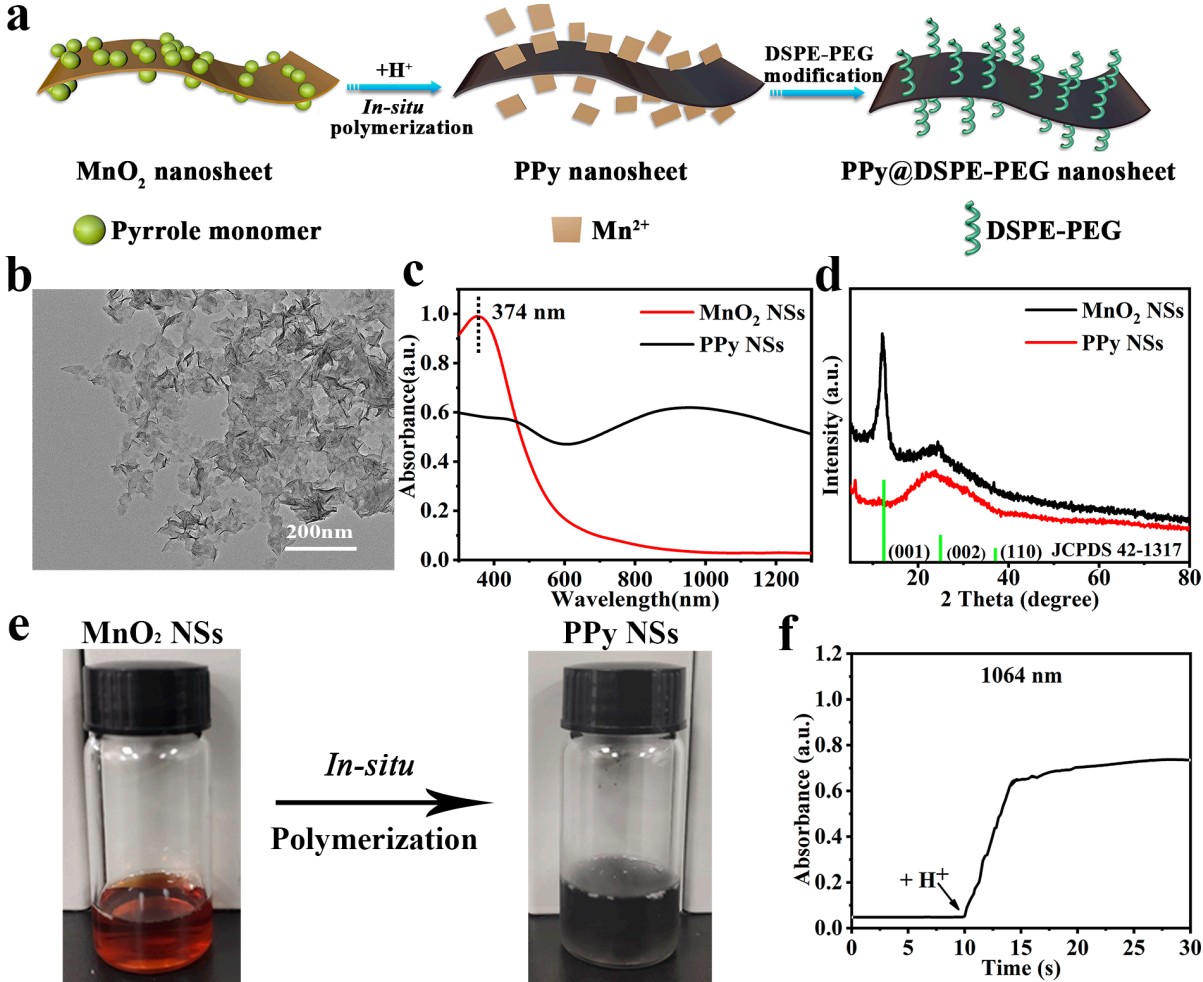

**Figure 2.** (**a**) Schematic illustration of the process for synthesizing PPy NSs by means of in situ oxidation of $MnO_2$ NSs. (**b**) TEM image of $MnO_2$ NSs. (**c**) UV–Vis–NIR absorption spectra of $MnO_2$ NSs and PPy NSs. (**d**) X-ray diffraction patterns of $MnO_2$ NSs and PPy NSs. (**e**) Photographs of $MnO_2$ NS (left) and PPy NS (right) aqueous solution. (**f**) Change of UV–Vis–NIR absorption value at 1064 nm during the synthesis of PPy NSs.

Notably, the $MnO_2$ NSs could be stably dispersed in DI water (Figure 2e, left) and showed no oxidation in neutral conditions. However, the oxidizability of the $MnO_2$ NSs was activated in an acidic solution, and the oxidative potential of $Mn^{4+}$ was high enough ($E_{Mn^{4+}/Mn^{2+}} = 1.224$ V) to initiate the polymerization of pyrrole (1.2 V vs. SCE). Therefore,

the color had no changes when pyrrole was mixed with the neutral dispersion of $MnO_2$ NSs, and pyrrole molecules were adsorbed on the surface of negatively charged $MnO_2$ NSs by electrostatic interaction (Figure S2, ESI†). However, after the addition of hydrochloric acid (HCl) solution, $MnO_2$ NSs were gradually dissolved and released $Mn^{4+}$ ions with strong oxidizability, which would lead to the polymerization of adsorbed pyrrole molecules on the surface of the $MnO_2$ NSs. This process was accompanied by a visible color change in the above mixed solution from brown–yellow to grey–black within only a few seconds (Figure 2e, right; video, ESI†).

In fact, the acidic condition was integral for initiating the polymerization of pyrrole, and the color change of the reaction solution was relatively rapid after the addition of HCl. We further tracked the real-time absorption of the reaction solution at 1064 nm to evaluate the progress of pyrrole polymerization (Figure 2f). It could be seen that the absorption curve of the solution showed no change before the addition of HCl, which further proved that the stable $MnO_2$ NSs could not oxidize pyrrole molecules. Only at the moment of adding HCl, the light absorption of the solution increased sharply, indicating that the polymerization of pyrrole had been initiated. After approximately 5 s, the absorption curve tended to be stable, indicating that the polymerization of pyrrole was almost achieved.

Due to the uncontrollability of the pyrrole polymerization, external limitation by a hard/soft template was necessary to obtain the PPy NSs. In this reaction, the PPy polymerization occurred on the surface of the $MnO_2$ NSs, which meant that the size and thickness of the PPy NSs were significantly affected by the templated $MnO_2$. Only by firstly obtaining high-quality ultrathin $MnO_2$ NSs can verified PPy NSs be obtained by means of a subsequent shape-copy reaction. Meanwhile, the lack of a Z-axis direction growth restriction, such as the FeOCl interlayer [34] or liquid–liquid interfaces [37], which had been used to fabricate PPy NSs in other reported methods, was not conducive to controlling the thickness of the PPy NSs. Comparing the thickness of PPy NSs prepared by different reported methods, it can also be seen that the thickness of the PPy NSs polymerized on the surface of the template (approximately 21 nm) [38] was significantly greater than that of PPy prepared by means of space-limited methods (1–2 nm) [34]. Therefore, the controllability and uniformity of the thickness of PPy NSs produced by our method were less successful than those reported in other methods with spatial constraints. However, by performing ultrasonic peeling, with sufficient energy and time, we could obtain large amounts of uniformly thin $MnO_2$ NSs (Figures 2b and S1a), which was helpful in overcoming the above-mentioned problem of thickness control in PPy NS fabrication.

Additionally, the reaction time also influenced the thickness of the PPy NSs. Although prolonging the reaction time was of benefit in improving the polymerization degree of PPy, it might also have caused the excessive growth of PPy along the Z-axis to increase the thickness. Therefore, it was better to shorten the reaction time so as to reduce the thickness of the PPy NSs. To investigate the effect of reaction time on PPy NS fabrication, the morphology and UV–Vis spectra of PPy NSs with different reaction times (1, 2, 5, 10, 30 min, and 24 h) were further characterized. The SEM and TEM images confirmed that prolonging the reaction time would increase the thickness of the PPy NSs. Moreover, too long a reaction time (24 h) could even lead to a serious accumulation of irregular PPy particles and a loss of nanosheet morphology (Figure S3a–c). The UV–Vis spectra of PPy NSs at different reaction time points (1, 2, 5, 10, 30 min, and 24 h) further proved that the reaction time of approximately 2 min was good enough for PPy polymerization. Prolonging the reaction time had no positive effect on improving the light absorption capacity of the PPy NSs in the NIR region (Figure S3d). Subsequently, the element distribution of the PPy NSs (reacted for 2 min) was detected by using EDS to test the purity of the product (Figure S4). The result suggested that the content of Mn ions was very low (0.32 wt.%), which demonstrated that the $MnO_2$ had almost been removed and the final product was pure PPy rather than $PPy/MnO_2$ composite. Therefore, it was sufficient to control the polymerization time of PY within 2 min after the HCl solution was added, which was significantly shorter than in the above-reported methods.

The UV–Vis–NIR absorption spectrum of the as-prepared PPy NSs displayed a strong broadband absorption in the whole NIR region, and the absorption peak at 374 nm belonging to $MnO_2$ disappeared (Figure 2c), which further confirmed the formation of PPy and the depletion of $MnO_2$. Compared with other reported PPy NSs [34], shortening the reaction time might weaken the absorption capacity of the PPy NSs in the NIR region, but the fabricated PPy NSs still demonstrated a good absorption property; in particular, the absorption intensity of PPy NSs at 1064 nm was obviously higher than that at 808 nm, and the extinction coefficients of PPy NSs at 808 nm and 1064 nm were 8.5 and 10.4 L $g^{-1}$ $cm^{-1}$, respectively (Figure S5, ESI†). The result essentially revealed that the fabricated PPy NSs had a stronger absorption capacity at 1064 nm than that of the reported PPy NPs [21,35]. The XRD spectrum of the PPy NSs only exhibited a broad diffraction peak at 26° (Figure 2d), indicating that the obtained PPy NSs were an amorphous polymer.

The morphology of the PPy NSs was examined by performing TEM and AFM, respectively (Figure 3a,b). It could be seen that the lateral size of the PPy NSs was approximately 100 nm, which corresponded to the result of the dynamic light scattering (DLS) measurement (Figure S6, ESI†). After the in situ polymerization of pyrrole on the surface of the $MnO_2$ NSs, the thickness of the PPy NSs was increased to 4~5 nm. Moreover, the increasing thickness led to greater rigidity in the PPy NSs than in the $MnO_2$ NSs; therefore, no more fold structures could be seen on the TEM image of the PPy NSs (Figure 3a).

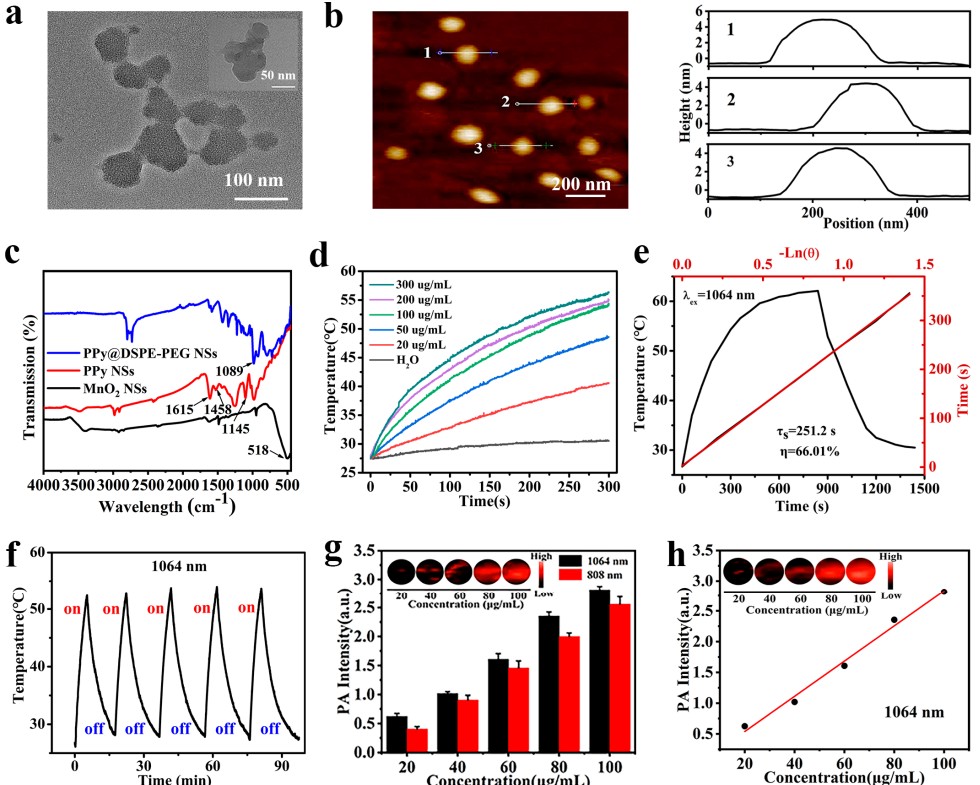

**Figure 3.** (**a**) TEM image of PPy NSs. (**b**) AFM image and corresponding height image of PPy NSs. (**c**) FTIR spectra of $MnO_2$ NSs, PPy NSs, and PPy@DSPE-PEG NSs. (**d**) The temperature-change curve of PPy@DSPE-PEG NSs in a heating/cooling cycle under 1064 nm laser and the linear relationship between time and -ln(θ) in the cooling process. (**e**) Photothermal curve of PPy@DSPE-PEG NSs with different concentrations under 1064 nm laser irradiation (1 W/cm$^2$, 5 min). (**f**) The temperature-change profiles of PPy@DSPE-PEG NS aqueous solution (100 μg/mL) under irradiation of 1064 nm laser (1 W/cm$^2$) for five laser on/off cycles. (**g**) Photoacoustic signal intensity of PPy@DSPE-PEG NSs with different concentrations at excitation wavelengths of 808/1064 nm. (**h**) Photoacoustic signal graphs of PPy@DSPE-PEG NSs with different concentrations at excitation wavelength of 1064 nm.

mPEG-DSPE was used as the stabilizing agent to enhance the dispersion stability and biocompatibility of the PPy NSs. After mPEG-DSPE modification, the size of the PPy@DSPE-PEG NSs was slightly increased to 133 nm (Figure S6, ESI†), and zeta potential changed from $-17.9$ mV of PPy NSs to $-26.1$ mV (Figure S2, ESI†). The increase in electronegativity gave the PPy@DSPE-PEG NSs long-term dispersion stability in various solvents without aggregation and size changes (Figure S7, ESI†). Moreover, the surface modification process had no effect on the absorption ability of the PPy@DSPE-PEG NSs in the NIR region (Figure S8, ESI†).

Figure 3c shows FTIR spectra of the PPy NSs before and after mPEG-DSPE modification, respectively. The peak at 518 $cm^{-1}$ was attributed to the Mn-O vibrations of $MnO_2$. For PPy NSs, peaks at 1615 and 1458 $cm^{-1}$ were related to the vibration of the pyrrole ring, and the peak at 1145 $cm^{-1}$ was assigned to the C-N tensile vibration, suggesting the successful formation of PPy [44]. Meanwhile, the disappearance of the characteristic peak of $MnO_2$ further indicated that $MnO_2$ was consumed during pyrrole polymerization. After modification, the C-O-C peak of PEG appeared at 1089 $cm^{-1}$, indicating the successful modification of PPy NSs by mPEG-DSPE.

### 3.2. Photothermal Properties of PPy@DSPE-PEG NSs

To evaluate the photothermal performance of PPy@DSPE-PEG NSs in the NIR-II region, PPy@DSPE-PEG NS solutions with various concentrations were exposed to a 1064 nm laser (1 $W/cm^2$) for 5 min, and the real-time temperature was acquired simultaneously by means of an infrared thermal camera. As shown in Figure 3d, the temperature of PPy@DSPE-PEG NS solutions (200 µg/mL) could rise to 55 °C within 5 min of irradiation by the 1064 nm laser, and the photothermal performance of the PPy@DSPE-PEG NSs was concentration-dependent. Notably, the PPy@DSPE-PEG NSs could even rise to 40.6 °C within 5 min of 1064 nm irradiation at a low concentration of 20 µg/mL, while the temperature of pure water was only increased by approximately 3 °C under the same conditions. Compared with the photothermal heating curve of the PPy@DSPE-PEG NSs under the 808 nm laser (1 $W/cm^2$, 5 min) (Figure S9a, ESI†), PPy@DSPE-PEG NSs could rise to a higher temperature when 1064 nm laser irradiation was performed with the same irradiation power density. Moreover, the photothermal conversion efficiency (PTCE) of the PPy@DSPE-PEG NSs could reach up to 66.01% at 1064 nm (Figure 3e), which was significantly higher than that (58.27%) at 808 nm (Figure S9b,c, ESI†), suggesting the PPy@DSPE-PEG NSs underwent excellent photothermal conversion in the NIR-II region. More importantly, since the 1064 nm laser has superior tissue-penetration ability than the 808 nm laser, and even in the presence of chicken breast with different thicknesses, the PPy@DSPE-PEG NS solution still exhibited effective photothermal performance under 1064 nm laser irradiation (Figure S10, ESI†).

The temperature-rise trend of the PPy@DSPE-PEG NS solution (100 µg/mL) had no obvious attenuation in five cycles of heating/cooling under 1064 nm laser irradiation (Figure 3f), proving the high photothermal stability of the PPy@DSPE-PEG NSs. In addition, the absorption spectra and morphology of the PPy@DSPE-PEG NSs before and after five-cycle laser irradiation were basically consistent, further indicating that the PPy@DSPE-PEG NSs possessed outstanding photostability (Figure S11, ESI†). Moreover, after immersion in the tumor environment simulation solution for 48 h, the morphology and light absorption performance of the PPy NSs showed no obvious reduction (Figure S12), which proved that the PPy had good structural stability. Therefore, after entering the complex tumor microenvironment, the PPy NSs can still maintain a stable structure for a certain time, which guarantees the following in vivo PTT and PA imaging applications.

### 3.3. Evaluation of In Vitro PA Imaging

Encouraged by the outstanding NIR absorption of PPy@DSPE-PEG NSs in the whole NIR region, the photoacoustic signals of PPy@DSPE-PEG NSs at the excitation wavelengths of 808 and 1064 nm were further measured. Obviously, the PPy@DSPE-PEG NSs displayed

higher PA signals at 1064 nm than at 808 nm at the same concentration (Figure 3g), which indicated that the PPy@DSPE-PEG NSs had NIR-II responsive PA imaging performance. Moreover, PA signals increased with increasing concentration and revealed linear concentration dependence (Figure 3h). These results indicated that the PPy@DSPE-PEG NSs could be regarded as an excellent PA contrast agent.

### 3.4. Cytotoxicity and In Vitro PTT Assay

Based on the excellent in vitro photothermal and PA imaging performances of PPy@DSPE-PEG NSs, they might be used as potential PATs for cancer treatment. Before that, we first need to examine the biocompatibility of PPy@DSPE-PEG NSs. As shown in Figure 4a, the viability of L929 and 4T1 cells was, in both cases, more than 90% even if the concentration of PPy@DSPE-PEG NSs was up to 400 µg/mL, indicating that the PPy@DSPE-PEG NSs had excellent biocompatibility and low cytotoxicity. Subsequently, the phototoxicity of the PPy@DSPE-PEG NSs against 4T1 cells was tested. It was noted that 808/1064 nm laser irradiation alone (1 W/cm$^2$, 10 min) did not cause photothermal damage on 4T1 cells. However, an enhanced concentration-dependent photocytotoxicity of the PPy@DSPE-PEG NSs on 4T1 cells was observed under 808/1064 nm laser irradiation conditions (1 W/cm$^2$, 10 min) (Figure 4b). Moreover, 1064 nm laser irradiation could introduce more severe photothermal damage on 4T1 cancer cells, and more than 90% of 4T1 cells were killed under the 1064 nm laser irradiation at a concentration of 50 µg/mL. The result of fluorescent live/death staining imaging visually displayed the photothermal lethality of PPy@DSPE-PEG NSs on 4T1 cells (Figure 4c). A significant red fluorescence from cell photothermal death could be observed when the concentration of PPy@DSPE-PEG NSs reached 30 µg/mL under the 808/1064 nm laser. All the above results revealed that the PPy@DSPE-PEG NSs could achieve an efficient PTT against 4T1 cells with NIR-II laser irradiation.

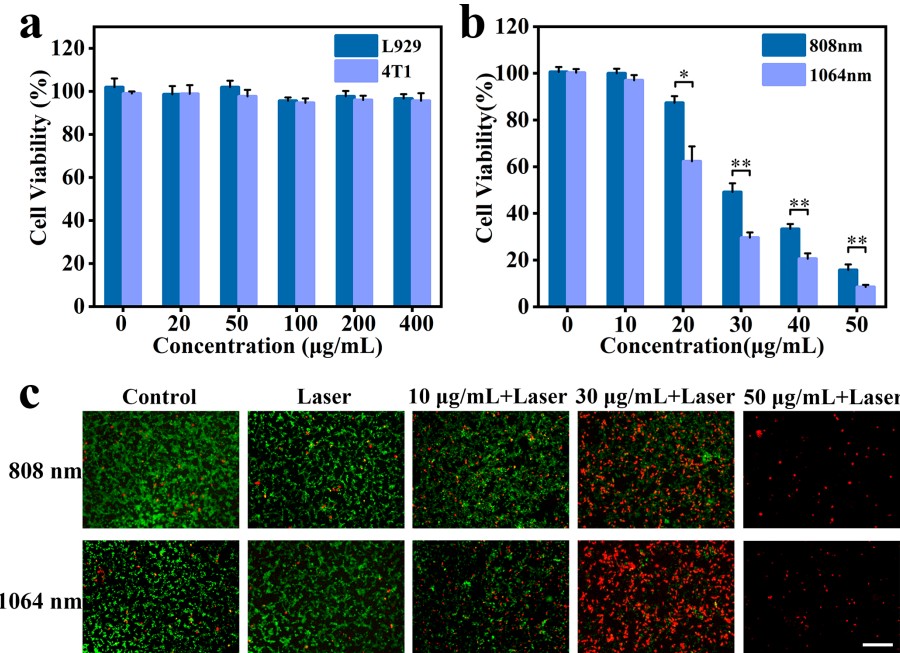

**Figure 4.** (**a**) Cytotoxicity of PPy@DSPE-PEG NSs at different concentrations against L929 cells and 4T1 cells. (**b**) Cytotoxicity of PPy@DSPE-PEG NSs at different concentrations against 4T1 cells treated with PPy@DSPE-PEG NSs under 808 nm/1064 nm laser (1 W/cm$^2$, 10 min). (**c**) Live/dead staining of 4T1 cells treated with PPy@DSPE-PEG NSs under 808 nm/1064 nm laser (1 W/cm$^2$, 10 min). (Scale bar = 200 µm). * represents $p < 0.05$. ** represents $p < 0.01$.

### 3.5. Evaluation of In Vivo PA Imaging

In vitro results stated that the PPy@DSPE-PEG NSs had excellent PA imaging and PTT performances against 4T1 cells under 1064 nm laser irradiation. We further studied

the in vivo PA imaging ability of PPy@DSPE-PEG NSs. The 4T1 tumor-bearing mice were injected intravenously with 100 μL of PPy@DSPE-PEG NS solution at the dose of 4 mg/kg; then, the PA intensity of the tumor site at 1064 nm was detected at 2, 4, 8, 12, and 24 h after injection, respectively. The maximum PA signal was reached at 8 h post-injection (Figure 5a,b), indicating PPy@DSPE-PEG NSs could be enriched to the tumor site through the EPR effect and produce a detectable in vivo NIR-II PA signal. Notably, the PA signal of the tumor site was still at a relatively effective response level 24 h after injection, indicating the long-term aggregation and retention of PPy@DSPE-PEG NSs in the tumor site (Figure S13, ESI†).

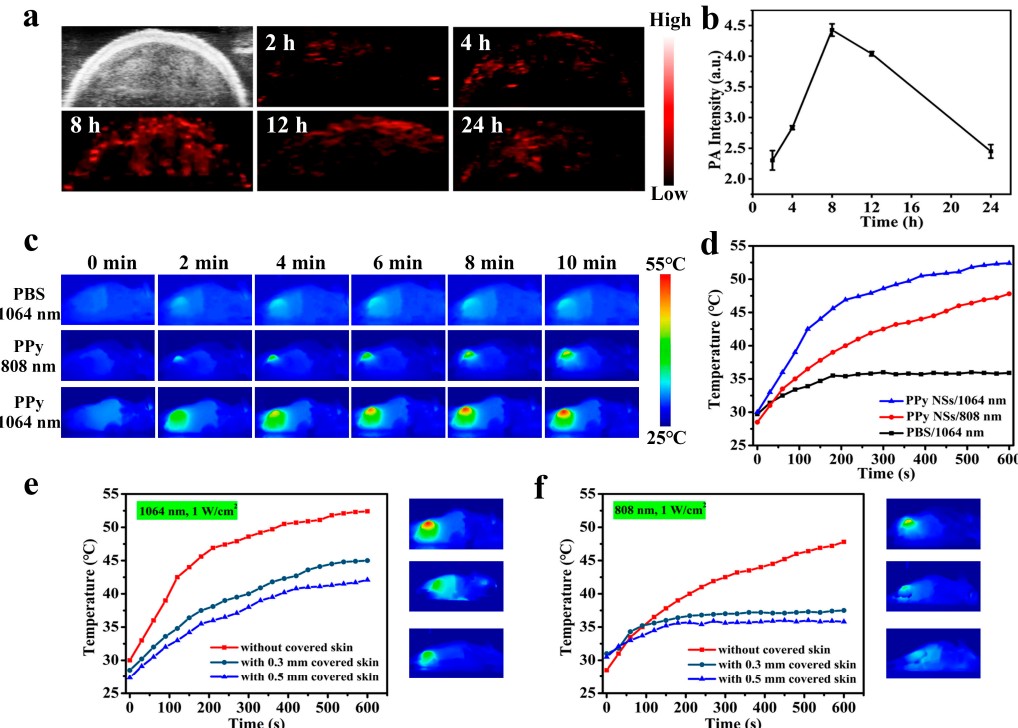

**Figure 5.** (**a**,**b**) Photoacoustic signal of tumor at 2, 4, 8, 12, and 24 h after intravenous injection of PPy@DSPE-PEG NSs (4 mg/kg) at 1064 nm excitation wavelength. (**c**) Infrared thermal imaging of 4T1 tumor-bearing mice after intravenous injection of PBS and PPy@DSPE-PEG NSs under 808/1064 nm laser irradiation (1 W/cm$^2$) within 10 min and (**d**) heating curve of irradiated tumor site. Heating curve of 4T1 tumor-bearing mice of the tumor site under 808 nm (**e**) and 1064 nm (**f**) laser irradiation (1 W/cm$^2$, 10 min) with/without covered skin.

### 3.6. In Vivo PTT

Inspired by the excellent in vitro PTT capacity of PPy@DSPE-PEG NSs, an in vivo photothermal effect procedure on 4T1-bearing mice was further carried out. The tumor site of 4T1 tumor-bearing mice was irradiated by 808 nm/1064 nm at the density of 1 W/cm$^2$ for 10 min after 8 h of PPy@DSPE-PEG NS injection. The real-time temperature of the tumor region was recorded by an infrared imaging camera. The temperature at the tumor could rise up to 53 °C (ΔT = 23 °C) and 47.8 °C (ΔT = 19.3 °C) within 10 min of laser irradiation using the 1064 and 808 nm laser, respectively, which was sufficient to cause photothermal damage to tumor tissue. Moreover, the temperature of the tumor site treated with 1064 nm laser irradiation showed a faster heating rate, as the real-time temperature of the tumor site could increase to 45 °C within 3 min of irradiation. In comparison, more than twice the irradiation time was needed to reach the same temperature with 808 nm laser irradiation (Figure 5c,d). Additionally, due to the better tissue-penetration ability of the 1064 nm laser, the temperature of the tumor site covered with chicken of different thicknesses (0.3 and 0.5 mm) could also reach up to above 42 °C after irradiation and lead to effective thermal

damage. In contrast, the temperature-rise tendency under 808 nm laser irradiation was inhibited by the covering tissue (Figure 5e,f).

The in vivo PTT effect of PPy@DSPE-PEG NSs against 4T1-bearing mice was appraised. The 4T1 tumor-bearing mice were randomly divided into six groups. Besides the control group, other mice were intravenously injected with PPy@DSPE-PEG NS solutions once only, with three consecutive-day photothermal therapy operations, while the tumor size and body weight measurements were observed continuously until the 14th day. After once only photothermal irradiation, obvious black spots of tissue thermal injury appeared at the tumor sites of the 808 nm/1064 nm direct irradiation groups and the group covered with 0.5 mm chicken and treated with 1064 nm irradiation, but no observable tissue thermal injury was observed in the group covered with 0.5 mm chicken and treated with 808 nm irradiation (Figure 6a).

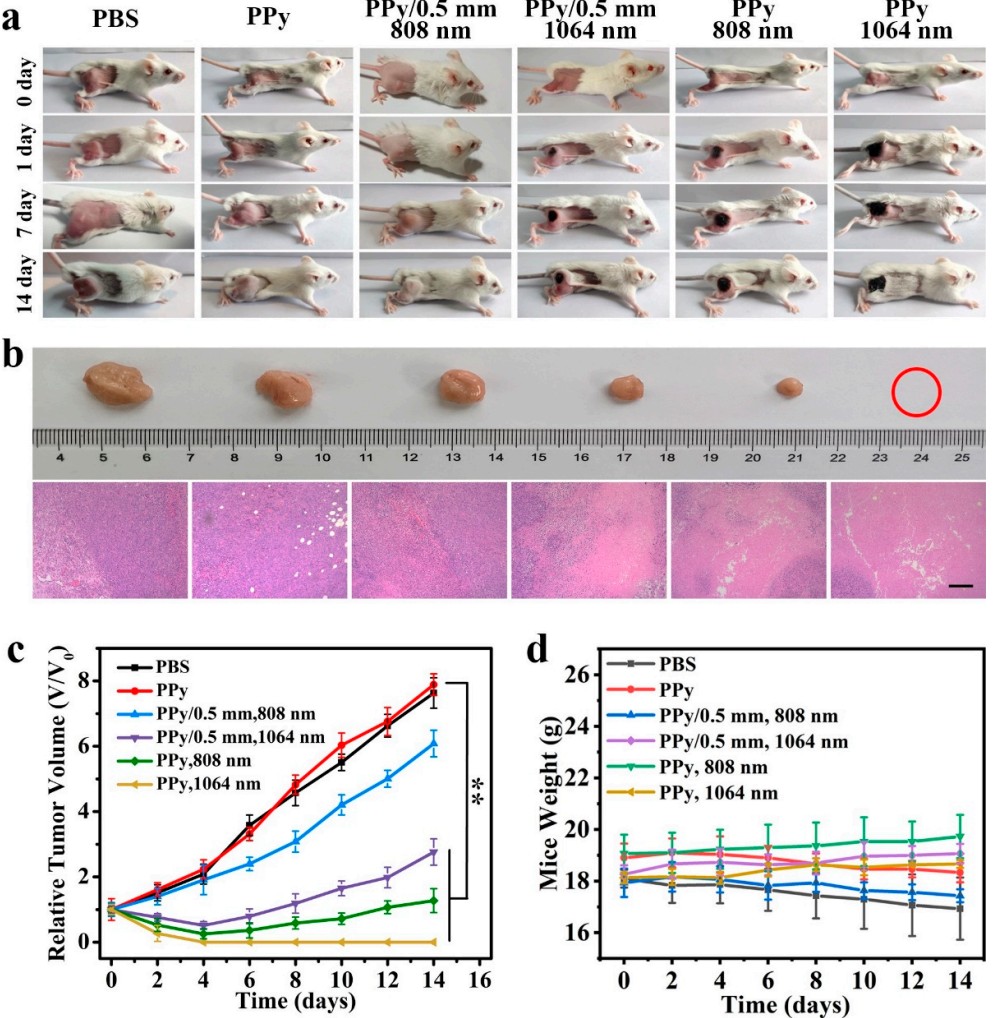

**Figure 6.** Mice images (**a**) of various groups after intravenous injection of different samples under corresponding laser irradiation (1 W/cm$^2$, 10 min). (**b**) The digital photos and H&E staining slices of tumors in different groups of 4T1 tumor-bearing mice. The H&E stainings were tested after once PTT and the digital photographs were taken after 14 days of treatment. (**c**) The relative tumor volume and (**d**) body weight of mice in different groups over 14 days. ** represents *p* < 0.01.

Obviously, the 1064 nm laser treated group showed the most effective photothermal ablation effect against 4T1 tumors. The H&E staining of tumor tissue after one irradiation treatment assay further revealed that the tumors in the above-mentioned three groups showed significant tissue necrosis. These results demonstrated that the PPy@DSPE-PEG NSs had a superior photothermal effect on deeper positioned tumors under 1064 nm

laser irradiation. The tumor size changes during the 14-day treatment and the images of tumor tissue in different groups clearly showed that only the PPy@DSPE-PEG NSs with 1064 nm laser irradiation exhibited effective photothermal efficacy in ablating tumor tissue thoroughly (Figure 6b,c). While the proliferation of tumors in the group irradiated with the 808 nm laser and the group covered with chicken and irradiated with the l064 nm laser was partly inhibited, the tumor tissue could not be completely ablated in a short time. These results further indicated that the PPy@DSPE-PEG NSs had an excellent PTT effect against 4T1 tumors. During the whole 14 days of PTT and the observation period, the body weight of the mice in all groups showed no obvious reduction (Figure 6d), confirming that the injected PPy@DSPE-PEG NSs had no toxicity to the physiological activities of the mice. The H&E staining images of major organs (heart, liver, spleen, lung, and kidney) in each group after the 14-day treatment displayed no noticeable damage, indicating that the PPy@DSPE-PEG NSs had excellent biocompatibility and long-term biosafety (Figure S14, ESI†).

## 4. Conclusions

In summary, we developed a fast and easy route to the fabrication of PPy NSs with a high-performance photothermal conversion capacity in the NIR-II region. The $MnO_2$ NSs as the 2D sacrificed template and oxidant directly guided and initiated the in situ polymerization of pyrrole on the $MnO_2$ NSs surface to form thin PPy NPs within a few minutes and without additional oxidant assistance. This method was faster than other reported methods for the preparation of 2D PPy structures and did not need additional template removal operations. Both in vitro and in vivo photothermal studies demonstrated that the prepared PPy NSs possessed good biocompatibility and exhibited excellent photothermal and photoacoustic properties under 1064 nm laser irradiation. They could achieve photoacoustic-guided photothermal treatment to effectively ablate 4T1 tumors under the 1064 nm laser. Therefore, PPy NSs are a promising PTA for cancer hyperthermia. Based on this, the PPy@DSPE-PEG nanosheets prepared in this paper can be further applied. The reaction conditions can be optimized to synthesize $MnO_2$-PPy composite nanosheets, which can use the Fenton-like effect of manganese ion to generate hydroxyl radicals to kill cancer cells and magnetic resonance imaging ability combined with the photoacoustic imaging ability of polypyrrole to guide the photothermal treatment of cancer under dual-mode imaging. Other chemotherapeutic agents, such as doxorubicin, can be combined on the nanosheets to achieve the bidirectional treatment of chemotherapy and phototherapy to improve the ability to perform tumor ablation. Photosensitizers, such as indocyanine green, can also be coated on nanosheets in combination with other optical treatment methods to enhance the effect of optical treatments. In conclusion, PPy nanosheets have a very wide application prospect in biomedical therapy.

**Supplementary Materials:** The following supporting information can be downloaded at: https://www.mdpi.com/article/10.3390/coatings13061037/s1, Figure S1: SEM (a) and AFM (b) images of $MnO_2$ NSs; Figure S2: Zeta potential of $MnO_2$ NSs, PPy NSs and PPy@DSPE-PEG NSs; Figure S3: Morphology of PPy NSs reacted with different time (a) 2 min, (b) 30 min, and (c) 24 h detected by SEM and TEM (insert pictures). (d) UV–Vis–NIR absorption spectra of PPy NSs under different synthetic time; Figure S4: EDS spectrum of PPy NSs; Figure S5: Extinction coefficient of PPy NSs at 1064 nm and 808 nm; Figure S6: DLS profiles of MnO2 NSs, PPy NSs, and PPy@DSPE-PEG NSs; Figure S7: DLS profiles of PPy@DSPE-PEG NSs in different buffer solutions; Figure S8: UV–Vis–NIR absorption spectra of PPy NSs with and without DSPE-PEG; Figure S9: (a) Photothermal heating curves of PPy@DSPE-PEG NSs with different concentrations under 808 nm laser irradiation (1 W/cm$^2$, 5 min). (b) Photothermal heating and cooling curves of PPy@DSPE-PEG NSs (100 μg/mL) under 808 nm and (c) the time constant for heat transfer from the system using a linear regression of the cooling profile; Figure S10: (a) Different thickness chicken breast pictures. (b) The temperature-change values of PPy@DSPE-PEG NSs (100 μg/mL) irradiated with 808 nm/1064 nm laser irradiation (1 W/cm$^2$, 5 min) under different thicknesses of covered skin; Figure S11: (a) UV–Vis–NIR absorption spectra of PPy@DSPE-PEG NSs before and after laser irradiation. (b) TEM image of PPy NSs after five-cycle laser irradiation; Figure S12: (a) Photothermal heating curves of PPy@DSPE-PEG NSs soaked and

unsoaked in the simulated solution of tumor environment (PBS with 5 mM GSH, pH = 5.0) under 1064 nm laser irradiation. (b) UV–Vis–NIR absorption spectra of PPy@DSPE-PEG NSs before and after being soaked in tumor microenvironment simulation fluid. (c) TEM image of PPy NSs after being soaked in the simulated solution of tumor environment; Figure S13: Pseudo-color pictures (a) and intensity (b) of photoacoustic signal intensity in major organs and tumor tissues after 24 h injection of PPy@DSPE-PEG NSs (4 mg/kg); Figure S14: H&E staining images of heart, liver, spleen, lung, and kidney at the end of the treatment cycle (14 day). (Scale bar = 200 μm). Video S1: Polymerization of PPy nanosheets.

**Author Contributions:** Methodology, Visualization, Validation, Investigation, Formal analysis, Writing—Original Draft, Y.X., J.X., and H.J.; Resources, Supervision, Y.Y.; Resources, Y.S. (Yuqing Shen), X.Z., and X.L.; Supervision, Y.S. (Yanan Sun); Supervision, Writing—Reviewing and Editing, W.S.; Conceptualization, Validation, Writing—Reviewing and Editing, Y.H.; Conceptualization, Supervision, Writing—Reviewing and Editing, D.G. All authors have read and agreed to the published version of the manuscript.

**Funding:** This work was financially supported by the National Natural Science Foundation of China (22272141, 22172132, 31870986), the Program for New Century Excellent Talents in University, the Natural Science Foundation of Fujian Province (2020J01036), the Foundation of Xiamen Science and Technology Bureau (3502Z20209206), the China Postdoctoral Science Foundation (2022M713855), and the Scientific Research Program of PLA (CWH17J030, BLB20J009).

**Institutional Review Board Statement:** The animal study protocol was approved by the Institutional Review Board of Xiamen University Laboratory Animal Center (XMULAC20170110, 2017-02-28).

**Informed Consent Statement:** Not applicable.

**Data Availability Statement:** Not applicable.

**Acknowledgments:** We acknowledge the analysis and testing center of Xiamen University.

**Conflicts of Interest:** The authors declare no conflict of interest.

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
