# Peer review of "Polypyrrole Nanosheets Prepared by Rapid In Situ Polymerization for NIR-II Photoacoustic-Guided Photothermal Tumor Therapy"

_coatings, doi:10.3390/coatings13061037_

Round 1

Reviewer 1 Report

In this manuscript 'Polypyrrole nanosheets prepared by rapid in-situ polymerization for NIR-II photoacoustic-guided photothermal tumor therapy' authors described a rapid and convenient in-situ polymerization strategy to fabricate polypyrrole nanosheets (PPy NSs) within a few minutes using manganese dioxide nanosheets (MnO2 NSs) as both the oxidant and self-sacrificing template. The fabricated PPy NSs exhibited excellent NIR-II absorption, which endowed its high photothermal conversion efficiency and photoacoustic (PA) imaging capability. Both in vitro and in vivo studies revealed that PPy NSs possess good biological safety, excellent PTT efficacy and PA imaging performances. Thus, the as-synthesized PPy NSs could achieve effectively PA imaging-guided photothermal tumor ablation. This work provides a novel and promising method for rapid preparation of PPy NSs without exogenous oxidants addition and subsequent template removal, which could be regarded as a potential photothermal agents (PTAs) to achieve the integration of cancer diagnosis and treatment.

This manuscript is in the scope of the journal is well written and interesting. Manuscript contributions to the field of biomedical chemistry. The manuscript can be published after some minor improvements and corrections:

Line 65 states that 'PPy as a typical conducting polymer has excellent biocompatibility', however, this statement should be supported by corresponding reference's (these sample articles are provided as a suggestion, but there may be others as well: Some Biocompatibility Aspects of Conducting Polymer Polypyrrole Evaluated with Bone Marrow-Derived Stem Cells. Colloids and Surfaces A: Physicochemical and Engineering Aspects 2014, 442, 152-156. // Evaluation of Cytotoxicity of Polypyrrole Nanoparticles Synthesized by Oxidative Polymerization. Journal of Hazardous Materials 2013 250-251, 167 –174.), which confirms good PPy biocompatibility with mammalian immune system and with living cells. In addition, these biocompatibility related issues should be discussed in chapter '3.4. Cytotoxicity and In Vitro PTT Assay' (Lines 365-385).

Rather interesting PPy synthesis route based on oxidation by MnO2 is addressed, but the discussion can be provided about some other PPy synthesis routes, that enable the formation of PPy-nanostructures within living cells, which are based on other oxidizers such as [Fe(CN )6]3- ions or other redox-abe species (these sample articles are provided as a suggestion, but there may be others as well: Cell-Assisted Synthesis of Conducting Polymer – Polypyrrole – for the Improvement of Electric Charge Transfer through Fungi Cell Wall. Colloids and Surfaces B-Biointerfaces 2019, 175, 671-679, etc.).

Data in figure 3h should be supported by error bars, if experiment was repeated.

Conclusions could be extended by some addition further perspective application of PPy nanostructures in biomedical treatment.

English language needs some minor revision.

The text appears to be well-written and structured, with clear and concise sentences that convey the intended message effectively. A few minor suggestions for improvement are:

- In line 44, it would be more appropriate to say "low invasiveness" instead of "minimal/non-invasiveness".

- Line 49 can be rephrased for clarity: "Developing appropriate PTAs for clinical photothermal therapy remains a major challenge to achieve highly efficient PTT."

- In line 53, it should say "transition metal dichalcogenides" instead of "transition metal dichalcogenide", since multiple materials are being referred to.

- Some of the references could be formatted in a consistent way (e.g., using either numbers or author-year citation style).

- Line 56 can be rephrased for clarity: "However, most PTAs are only responsive to light in the NIR-I region, which has weaker tissue penetration depth and higher tissue scattering and reabsorption, as well as a lower maximum allowable exposure threshold compared to the NIR-II optical window [27, 28]."

- In line 60, it should say "the light absorption properties of conjugated polymers" instead of "the light absorption properties of conjugated polymers".

- In line 65, it would be more appropriate to say "there have been several studies" instead of "there have several studies".

- Line 73 is missing the word "for": "But the utmost problem in the preparation of PPy nanomaterials is poor controllability for polymerization and morphology."

- Line 75 can be rephrased for clarity: "Cai et al. reported efficient photothermal performance from hexagonal polypyrrole (PPy) NSs that were prepared by complex and fine interfacial polymerization [34]."

- It might help to include a clearer statement about what PA imaging is or means before it is first mentioned in sentence 88.

The overall text appears to be well-written and structured.

Author Response

Manuscript ID: coatings-2404651

Type: research article

Coatings

May. 29, 2023

Dear Editor and Reviewers,

We appreciate editor and reviewers very much for their positive and constructive comments and suggestions on our manuscript entitled “Polypyrrole nanosheets prepared by rapid in-situ polymerization for NIR-II photoacoustic-guided photothermal tumor therapy”. We have further finished the content and references revisions, as well as our manuscript accordingly and answered the issues raised by the reviewers point by point at the end of this letter. Besides, we also carefully checked some statements, edition and calculation errors. The corresponding content has been edited by highlight in revised manuscript.

Thank you for your kind consideration.

Sincerely yours,

Prof. Dongtao Ge

College of Materials, Xiamen University, China

Point-by-point response to reviewers’ comments

  1. Line 65 states that 'PPy as a typical conducting polymer has excellent biocompatibility', however, this statement should be supported by corresponding reference's (these sample articles are provided as a suggestion, but there may be others as well: Some Biocompatibility Aspects of Conducting Polymer Polypyrrole Evaluated with Bone Marrow-Derived Stem Cells. Colloids and Surfaces A: Physicochemical and Engineering Aspects 2014, 442, 152-156. // Evaluation of Cytotoxicity of Polypyrrole Nanoparticles Synthesized by Oxidative Polymerization. Journal of Hazardous Materials 2013 250-251, 167 –174.), which confirms good PPy biocompatibility with mammalian immune system and with living cells. In addition, these biocompatibility related issues should be discussed in chapter '3.4. Cytotoxicity and In Vitro PTT Assay' (Lines 365-385).

Response: Thanks for your comment. The corresponding references have been inserted in line 65. Many studies have shown that PPy has excellent biocompatibility as stated in the references we cited in line 44 and line 65. And we have also discussed about good biocompatibility of the PPy nanosheets with living cells and with living mammals in chapter 3.4 and chapter 3.6.

  1. Rather interesting PPy synthesis route based on oxidation by MnO2is addressed, but the discussion can be provided about some other PPy synthesis routes, that enable the formation of PPy-nanostructures within living cells, which are based on other oxidizers such as [Fe(CN )6]3-ions or other redox-abe species (these sample articles are provided as a suggestion, but there may be others as well: Cell-Assisted Synthesis of Conducting Polymer – Polypyrrole – for the Improvement of Electric Charge Transfer through Fungi Cell Wall. Colloids and Surfaces B-Biointerfaces 2019, 175, 671-679, etc.)

Response: Thanks for your rigorously comment. We have talked about some other researches about PPy synthesis routes from lines 73 to 81, and have inserted the reference in line 79.

  1. Data in figure 3h should be supported by error bars, if experiment was repeated.

Response: Thanks for your comment. Data in figure 3h shows photoacoustic signal of PPy@DSPE-PEG NSs with different concentrations at excitation wavelengths of 1064 nm, and the error bars have already been reflected in figure 3g.

  1. Conclusions could be extended by some addition further perspective application of PPy nanostructures in biomedical treatment.

Response: Thanks for your useful comment. We have already discussed more about the biomedical applications of PPy nanostructures in the conclusion.

  1. In line 44, it would be more appropriate to say "low invasiveness" instead of "minimal/non-invasiveness".

Response: Thanks for your careful comment. We have already changed "low invasiveness" to "minimal/non-invasiveness".

  1. Line 49 can be rephrased for clarity: "Developing appropriate PTAs for clinical photothermal therapy remains a major challenge to achieve highly efficient PTT."

Response: Thanks for your earnest comment. We have revised as suggested.

  1. In line 53, it should say "transition metal dichalcogenides" instead of "transition metal dichalcogenide", since multiple materials are being referred to.

Response: Thank you for your responsible comment. It has already been written as "transition metal dichalcogenides" instead of "transition metal dichalcogenide".

  1. Some of the references could be formatted in a consistent way (e.g., using either numbers or author-year citation style).

Response: Thank you for your comment. We have unified the forms of references.

  1. Line 56 can be rephrased for clarity: "However, most PTAs are only responsive to light in the NIR-I region, which has weaker tissue penetration depth and higher tissue scattering and reabsorption, as well as a lower maximum allowable exposure threshold compared to the NIR-II optical window [27, 28]."

Response: Thank you for your useful comment. We have already modified as requested.

  1. In line 60, it should say "the light absorption properties of conjugated polymers" instead of "the light absorption property of conjugated polymers".

Response: Thanks for your careful comment. We have already replaced "the light absorption properties of conjugated polymers" with "the light absorption properties of conjugated polymers".

  1. In line 65, it would be more appropriate to say "there have been several studies" instead of "there have several studies".

Response: Thanks for your useful comment. We have already amended as requested.

  1. Line 73 is missing the word "for": "But the utmost problem in the preparation of PPy nanomaterials is poor controllability for polymerization and morphology."

Response: Thanks for your aborative comment. This sentence has been corrected in line 73.

  1. Line 75 can be rephrased for clarity: " PPy) NSs that were prepared by complex and fine interfacial polymerization [34]."

Response: Thanks for your thoughtful comment. It has been revised as suggested.

  1. It might help to include a clearer statement about what PA imaging is or means before it is first mentioned in sentence 88.

Response: Thanks for your useful comment. Photoacoustic imaging (PAI) is a new imaging mode, which can obtain the optical con-trast in the tissue. Polypyrrole has been shown to have certain photoacoustic imaging properties. And NIR-II has better tissue penetration than NIR-I. Therefore, high-resolution and high-contrast photoacoustic imaging images of tumor sites can be obtained under the irradiation of the NIR-II laser. So as to effectively improve the effect of photothermal treatment. We have added a clear statement about PA imaging in line 67, and have inserted corresponding references as well.

Reviewer 2 Report

30% similarity may acceptable, however, in experimental section, it is very high amount of similarity. Should be checked and re-written. Moreover, English check is needed. I could not see in vivo side effects. Any study for this? Please identify. Moreover, comparison with the literature and yield amount is missing. Please add them

Should be improved a little bit.

Author Response

Manuscript ID: coatings-2404651

Type: research article

Coatings

May. 29, 2023

Dear Editor and Reviewers,

We appreciate editor and reviewers very much for their positive and constructive comments and suggestions on our manuscript entitled “Polypyrrole nanosheets prepared by rapid in-situ polymerization for NIR-II photoacoustic-guided photothermal tumor therapy”. We have further finished the content and references revisions, as well as our manuscript accordingly and answered the issues raised by the reviewers point by point at the end of this letter. Besides, we also carefully checked some statements, edition and calculation errors. The corresponding content has been edited by highlight in revised manuscript.

Thank you for your kind consideration.

Sincerely yours,

Prof. Dongtao Ge

College of Materials, Xiamen University, China

Point-by-point response to reviewers’ comments

1.30% similarity may acceptable, however, in experimental section, it is very high amount of similarity. Should be checked and re-written. Moreover, English check is needed.

Response: Thanks for your thoughtful comment. All experimental operations in this paper were general and mature, and were written by ourselves. As we can see in the pictures below, some of the correspondence addresses, professional terms, regular experimental operations, drug and instrument names do show some duplication with other literature, which is inevitable. And we have rewritten part of the experimental section and checked the writing of English. Pictures shown in our rechecking report are as follows:

  1. I could not see in vivo side effects. Any study for this? Please identify.

Response: Thanks for your aborative comment. There are several studies showing that PPy has excellent biocompatibility, low cytotoxicity, and few side effects in vivo, as the references state we cite in line 44 and line 65. Our subsequent cell experiments in chapter 3.4 and animal experiments in chapter 3.6 also proved that the PPy nanosheets has no toxicity. As shown in Figure 4a, the viability of L929 and 4T1 cells were both more than 90% after 24 h of incubation even if the concentration of PPy@DSPE-PEG NSs was up to 400 μg/mL. And the body weight of mice had no obvious drop during the whole 14-day of PTT and observation period as shown in Figure 6d. Meanwhile, H&E staining images of main organs (heart, liver, spleen, lung, and kidney) in Figure S14 also displayed no noticeable damages, indicating the PPy@DSPE-PEG NSs had excellent biocompatibility and long-term biosafety.

  1. Moreover, comparison with the literature and yield amount is missing. Please add them.

Response: Thanks for your thoughtful comment. We have compared with other literature. For example, we compared different synthesis methods of PPy in lines 81 to 94 to illustrate that in situ oxidation template method of forming PPy nanosheets is very convenient. As for yield amount, it is directly related to the amount of MnO2 involved in the reaction. More MnO2 added, more products are obtained. Thus, it is not very necessary to calculate the yield amount. Rather than the yield, the morphology, stability and light absorption of nanosheets should be more concerned. The relevant tests in Figure 2 and Figure 3 confirmed that the nanosheets synthesized had uniform size, good stability and excellent light absorption, and had high photothermal conversion efficiency in NIR-I and NIR-II, which could also be shown in our subsequent tumor treatment.

Reviewer 3 Report

 1- line 118: Finally, 10 mg of MnO2 was ground and dispersed in 20 mL ultrapure water and ultra-sonicated (power: 480 W) for 12 h to strip MnO2 NSs.(12 hrs by sonication is very high, please confirm, and how many rounds/second)

2- On the 14th day of treat- 201 ment, all mice were sacrificed and major organs (heart, liver, spleen, lung and kidney) and tumors were harvested for H&E staining to analyze the biological safety.  ( sacrificed or anesthetized, please add the certificate (no) of the ethical committee)

3- please add statistical analysis in the methods 

4- lignad of figure 4a, 4b. changed to

 fig 4a Cytotoxicity of PPy@DSPE-PEG NSs at different concentrations against L929 cells  and  4T1 cells (a)

Fig 4b: Cytotoxicity of PPy@DSPE-PEG NSs at different concentrations against 4T1 cells s treated with PPy@DSPE-PEG NSs under 808 384 nm/1064 nm laser (1 W/cm2 , 10 min)

Author Response

Manuscript ID: coatings-2404651

Type: research article

Coatings

May. 29, 2023

Dear Editor and Reviewers,

We appreciate editor and reviewers very much for their positive and constructive comments and suggestions on our manuscript entitled “Polypyrrole nanosheets prepared by rapid in-situ polymerization for NIR-II photoacoustic-guided photothermal tumor therapy”. We have further finished the content and references revisions, as well as our manuscript accordingly and answered the issues raised by the reviewers point by point at the end of this letter. Besides, we also carefully checked some statements, edition and calculation errors. The corresponding content has been edited by highlight in revised manuscript.

Thank you for your kind consideration.

Sincerely yours,

Prof. Dongtao Ge

College of Materials, Xiamen University, China

Point-by-point response to reviewers’ comments

  1. Line 118: Finally, 10 mg of MnO2 was ground and dispersed in 20 mL ultrapure water and ultra-sonicated (power: 480 W) for 12 h to strip MnO2 NSs.(12 hrs by sonication is very high, please confirm, and how many rounds/second).

Response: Thanks for your comment. Synthesis of MnO2 nanosheets was performed as described in references 41 and 42, which was reasonable and feasible. 10 mg MnO2 was dispersed in 20 mL distilled water with ultrasonication for 12 h. Then, the solution was centrifuged at 2000 r/min, and and the MnO2 NSs in the supernatant was collected for further use.

  1. On the 14th day of treat- 201 ment, all mice were sacrificed and major organs (heart, liver, spleen, lung and kidney) and tumors were harvested for H&E staining to analyze the biological safety. ( sacrificed or anesthetized, please add the certificate (no) of the ethical committee).

Response: Thanks for your aborative comment. All animal experiments were conducted in accordance with the guidelines and policies approved by the Experimental Animal Ethics Guidance Management Committee of Xiamen University, and the ethical number was XMULAC20170110, which was approved on February 28, 2017. The ethical number has been added in line 514.

  1. Please add statistical analysis in the methods.

Response: Thanks for your comment. We have added the formula of photothermal conversion efficiency of PPy@DSPE-PEG NSs in line 166. And all other data were processed and plotted in the usual way.

  1. Lignad of figure 4a, 4b. changed to

 Fig 4a: Cytotoxicity of PPy@DSPE-PEG NSs at different concentrations against L929 cells and 4T1 cells.

Fig 4b: Cytotoxicity of PPy@DSPE-PEG NSs at different concentrations against 4T1 cells s treated with PPy@DSPE-PEG NSs under 808 nm/1064 nm laser (1 W/cm2 , 10 min).

Response: Thanks for your useful comment. We have already revised as suggested.